# Biofilm heterogeneity-adaptive photoredox catalysis enables red light-triggered nitric oxide release for combating drug-resistant infections

Jian Cheng[1], Guihai Gan[2], Shaoqiu Zheng[2], Guoying Zhang[2], Chen Zhu [1] ✉, Shiyong Liu [2] ✉ & Jinming Hu [2] ✉

The formation of biofilms is closely associated with persistent and chronic infections, and physiological heterogeneity such as pH and oxygen gradients renders biofilms highly resistant to conventional antibiotics. To date, effectively treating biofilm infections remains a significant challenge. Herein, we report the fabrication of micellar nanoparticles adapted to heterogeneous biofilm microenvironments, enabling nitric oxide (NO) release through two distinct photoredox catalysis mechanisms. The key design feature involves the use of tertiary amine (TA) moieties, which function as sacrificial agents to avoid the quenching of photocatalysts under normoxic and neutral pH conditions and proton acceptors at acidic pH to allow deep biofilm penetration. This biofilm-adaptive NO-releasing platform shows excellent antibiofilm activity against ciprofloxacin-resistant *Pseudomonas aeruginosa* (CRPA) biofilms both in vitro and in a mouse skin infection model, providing a strategy for combating biofilm heterogeneity and biofilm-related infections.

Photoredox catalysis has emerged as a powerful tool in organic and polymer synthesis, characterized by its ability to initiate challenging and unconventional bond-forming/cleaving reactions[1–3]. By harnessing the precise spatiotemporal control of light irradiation and mild reaction conditions, photoredox catalysis has recently been successfully operated in living systems for protein proximity labeling[4], cancer therapy[5,6], antibacterial applications[7–9], and more. To mitigate phototoxicity and increase tissue penetration of incident light, some recent advancements have been achieved in the development of red or near-infrared red (NIR) light-driven photoredox catalysis reactions. Notably, while many photoredox catalysis reactions proceed smoothly in non-living systems, they can be deactivated by complex biological microenvironments through various pathways[10]. For example, heterogeneously distributed oxygen and biomolecules can greatly quench

photocatalysts (PCs). Moreover, intramolecular energy/electron transfer from PCs to the target substances can be impeded due to their distinct biodistributions. Therefore, developing photoredox catalysis platforms adapted to highly heterogeneous biological microenvironments, which is critical for achieving targeted functions, remains a significant challenge.

As a ubiquitous lifestyle, bacteria in nature spontaneously form biofilms to survive in hostile environments[11,12]. Bacterial biofilms are clusters of microorganisms composed of extracellular polymeric substances (EPS), which contain polysaccharides, proteins, glycoproteins, nucleic acids, and more[13–15]. Once formed, bacterial biofilms can evade immune surveillance and develop resistance to antibiotics, representing a more severe threat to human society than planktonic bacteria[12,16]. Indeed, approximately 80% of persistent and intractable

[1]Department of Orthopedics, The First Affiliated Hospital of University of Science and Technology of China (USTC), Division of Life Sciences and Medicine, University of Science and Technology of China, Hefei, Anhui Province 230001, China. [2]Department of Pharmacy, The First Affiliated Hospital of University of Science and Technology of China (USTC), and Key Laboratory of Precision and Intelligent Chemistry, Department of Polymer Science and Engineering, University of Science and Technology of China, Hefei, Anhui Province 230026, China. ✉e-mail: zhuchena@ustc.edu.cn; sliu@ustc.edu.cn; jmhu@ustc.edu.cn

infections are closely related to biofilm formation[17]. It is widely appreciated that bacterial cells within biofilms have different physiological states and distinct metabolic pathways, resulting in physiological heterogeneity in biofilms such as oxygen and pH gradients. The highly heterogeneous microenvironment within biofilms, in turn, inhibits the penetration of antibiotics and increases drug resistance, making it rather challenging to treat biofilm infections[18–20].

To combat biofilm infections, several approaches have been extensively studied, including the use of antimicrobial peptides[21,22], bacteriophages[17,23], biofilm-dispersing enzymes[24], and nitric oxide (NO)[25–27]. Among these, NO, an endogenous free radical with the ability to simultaneously eradicate biofilms and kill planktonic bacteria, has received increasing attention[28,29]. To increase therapeutic efficacy and reduce side effects, a variety of NO-releasing platforms have been developed to achieve targeted and sustained delivery of NO, exhibiting antibacterial and antibiofilm activities[30–32]. In this regard, we recently developed NO-releasing micellar nanoparticles that can be selectively activated by a photoredox catalysis mechanism under hypoxic biological environments[8], and oxygen-tolerant photoredox catalysis can further be achieved by introducing reactive oxygen species (ROS)-scavenging agents[7,9]. Although these NO-releasing micelles efficiently killed planktonic bacteria under normoxic conditions, they showed insufficient antibiofilm activity due to poor biofilm penetration and compromised photoredox catalysis in heterogeneous biofilm microenvironments.

In this study, we fabricated micellar nanoparticles by co-assembling diblock copolymers bearing NO-releasing moieties/Pd-based photocatalyst (PC) and tertiary amine (TA) residues/Pd-based

PC. These micellar nanoparticles can adapt to biofilm microenvironments and release NO upon red light irradiation, employing distinct photoredox catalysis mechanisms (Fig. 1). The key design feature involved incorporating TA moieties, which served as both ROS-scavenging agents and proton acceptors. Specifically, under high pH and normoxic conditions such as those found in the periphery of biofilms, the deprotonated TA moieties allowed them to scavenge singlet oxygen ($^1O_2$) to avoid oxygen quenching of the Pd-based PC and facilitate NO release. On the other hand, in response to the local acidic pH in the inner layer of biofilms, these TA moieties became positively charged, enhancing the penetration of the resulting micellar nanoparticles into the deeper layers of biofilms. Within the inner layers of biofilms, photoredox catalysis occurred under hypoxic conditions, leading to subsequent NO release. We demonstrated that biofilm microenvironment-adaptive NO-releasing micelles alone can release NO in heterogeneous biofilm microenvironments, effectively eradicating ciprofloxacin-resistant *Pseudomonas aeruginosa* (CRPA) biofilms both in vitro and in a skin infection mouse model.

## Results

### Red light-mediated NO release through photoredox catalysis

To achieve antibiofilm activity, we initially carefully designed and screened NO donors that can potentially be activated by photoredox catalysis. *N*-nitrosoamine derivatives have been widely used as photoresponsive NO donors[33]. We recently found that coumarin- and nitrobenzofurazan (NBD)-based *N*-nitrosoamine donors containing appropriate electron-withdrawing groups could be selectively activated by photoredox catalysis mechanisms[7,8]. To maximize the NO

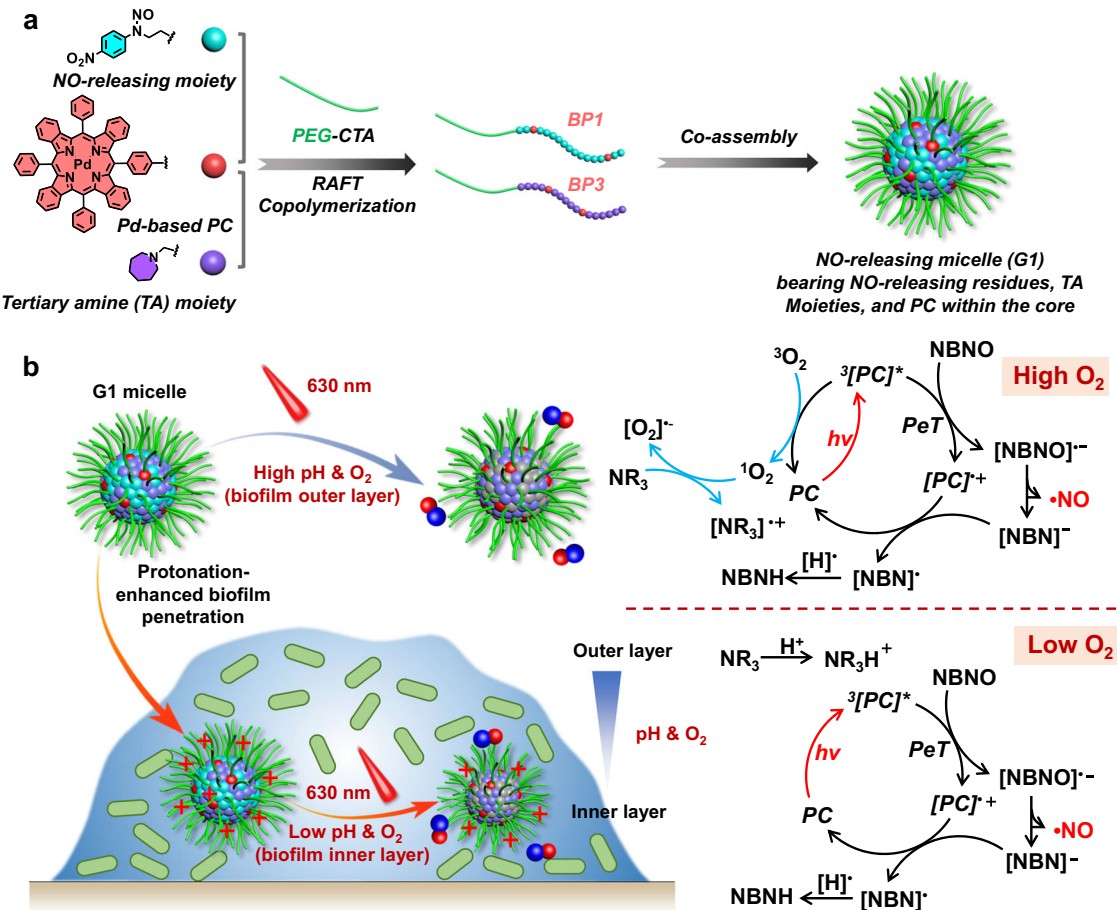

**Fig. 1 | Fabrication of biofilm microenvironment-adaptive photoredox catalysis platform. a** Schematic illustration of the co-assembly of BP1 and BP3 copolymers with the formation of NO-releasing G1 micelles with NO-releasing residues, tertiary amine (TA) moieties, and Pd-based PC within the cores. **b** Plausible mechanisms of red light-mediated NO release from G1 micelles in heterogeneous biofilms.

loading contents, aniline derivatives were used to synthesize corresponding N-nitrosoamine-based NO donors and the NO loading contents were calculated (14.2% for NBNO, 15.3% for MBNO, and 18% for BNO; Supplementary Fig. 1). The chemical structures of these potential NO donors were characterized by nuclear magnetic resonance (NMR) and high-resolution mass spectrometry (HRMS) (Supplementary Figs. 2–7). NBNO, MBNO, and BNO donors did not exhibit absorbance above 400 nm (Supplementary Fig. 8) and photo-mediated NO release needed to be conducted under UV light irradiation[34]. We tested whether these NO donors could be activated via a photoredox catalysis process. Interestingly, a mixture of NBNO (100 μM) and PdTPTBP (5 μM) in DMSO showed a remarkable absorbance decrease at 318 nm and an increase at 398 nm under mild 630 nm irradiation (39 mW/cm²), whereas MBNO and BNO cannot be activated under the same conditions. In addition, no absorbance changes were observed for the NBNO/PdTPTBP mixture under dark conditions (Supplementary Fig. 9).

Electron spin resonance (ESR) spectra using 2-phenyl-4,4,5,5-tetramethylimidazoline-1-oxyl-3-oxide (PTIO) as the spin-trapping agent revealed the release of NO radicals for the mixture of NBNO/PdTPTBP under 630 nm light irradiation (Supplementary Fig. 10). The photolysis quantum yield was calculated to be 0.76% using Reinecke's salt actinometry (Supplementary Fig. 11)[35]. Notably, the NO release rates can be readily adjusted by the amounts of PdTPTBP and a high PC content led to faster NO release under identical irradiation conditions (Supplementary Fig. 12 and Supplementary Table 1). Moreover, we found the exclusive formation of the NBNHM precursor from the NBNOM monomer under 630 nm light irradiation, while UV 365 nm light irradiation resulted in the formation of side products, as evidenced by HPLC analysis (Supplementary Figs. 13 and 14). This result revealed that the low photon energy of red light efficiently decreased the side reactions and contributed to the clean conversion of NO donors, which was quite favorable for understanding their biological functions.

To clarify the underlying mechanism of red light-mediated NO release, we first calculated the LUMO levels of PdTPTBP, NBNO, MBNO, and BNO, which were determined to be −2.60, −3.25, −1.67, and −1.77 eV, respectively (Fig. 2a). The excited redox potential of $E_{1/2}$(PdTPTBP$^{·+}$/PdTPTBP$^*$) was measured using cyclic voltammetry and determined to be −0.77 V (potential vs Ag/AgCl electrode in DMF), while the reduction potential of NBNO was calculated to be −0.75 V (Supplementary Fig. 15). In addition, the phosphorescence intensities and lifetimes of PdTPTBP were gradually quenched with increasing concentrations of NBNO. The Stern-Volmer constant ($K_{sv}$) and quenching constant ($K_q$) were calculated to be 124 M$^{-1}$ and 7.33 × 10$^5$ M$^{-1}$s$^{-1}$, respectively (Supplementary Fig. 16). Nanosecond transient absorption spectra also confirmed the quenching process, revealing a decrease in the excited lifetime from -144 to -12.8 μs (Supplementary Fig. 17). Collectively, NBNO can be reduced by the excited state of PdTPTBP through a photoinduced electron transfer (PeT) process, while the reduction of MBNO and BNO was less effective, in good agreement with the UV-vis absorbance spectra (Supplementary Fig. 9).

Additionally, we conducted further tests on Ru(bpy)₃, eosin Y, and rhodamine B (RhB) as potential PCs to assess the generality of NBNO activation through a photoredox catalysis mechanism. Ru(bpy)₃ (480 nm) and eosin Y (560 nm) showed good catalytic activity toward NBNO, while RhB (560 nm) showed little activity (Supplementary Figs. 18–20). Thus, activation of the NBNO donor can be achieved with different wavelengths. We chose PdTPTBP for the following studies because of its long-absorbing wavelength (e.g., 630 nm), which was beneficial for biomedical applications due to its decreased phototoxicity and increased tissue penetration[4,36].

Marcus theory was used to calculate the activation Gibbs free energy barrier of electron transfer (ET) from [PdTPTBP$^*$] to NBNO, which was determined to be $\triangle G^\dagger = 0.51$ kcal/mol, while much higher energy barriers were observed for MBNO (21.03 kcal/mol) and BNO (18.66 kcal/mol) (Fig. 2b). Building on the above results, we proposed the following activation mechanism for NO release. Under red light irradiation and hypoxic conditions, the triplet state of PdTPTBP reduced NBNO via an ET process with the formation of [NBNO]$^{·-}$ and [PdTPTBP]$^{·+}$. The [NBNO]$^{·-}$ intermediate spontaneously released NO radicals and formed [NBN]$^-$, while [NBN]$^-$ was further oxidized by [PdTPTBP]$^{·+}$ to [NBN]$^·$. The PC then returned to the ground state and [NBN]$^·$ was further transformed into NBNH following the abstraction of [$^·$H] from the surrounding medium (Fig. 1b and Supplementary Fig. 21a). It is worth noting that more complex reaction mechanisms could be involved under true biological conditions. For example, [NBN]$^-$ could be readily protonated under aqueous conditions, especially in pathological acidic conditions, and [PdTPTBP]$^{·+}$ could likely be oxidized by other biological oxidative agents as well.

Notably, oxygen can greatly retard and even completely inhibit photoredox catalysis reactions due to the quenching of PCs[5,6]. We then evaluated the oxygen influence on the photoredox catalysis process between PdTPTBP and NBNO in aerated DMF; unlike DMSO, DMF cannot efficiently scavenge reactive oxygen species (ROS)[37]. Remarkable retardation and PC bleaching were observed for NBNO activation by PdTPTBP, resulting in reduced NO release (Supplementary Fig. 22). Therefore, the activation of NBNO by PdTPTBP was greatly hampered by oxygen.

To overcome this limitation, we sought to identify appropriate sacrificial agents capable of mitigating the adverse effect of oxygen. Tertiary amine (TA) derivatives have previously been used as additives to increase oxygen tolerance in photoredox catalysis reactions[38,39]. Intriguingly, the addition of TA-containing methacrylate monomers (e.g., DEA, DPA, and C7A) can greatly accelerate the photoredox catalysis reactions and mitigate PC bleaching in aerated DMF (Supplementary Fig. 23). Quantitative analysis revealed that the activity of the three monomers was in the following order: C7A ≈ DEA > DPA. The lower activity of DPA was likely ascribed to its larger steric hindrance[40]. This result clearly demonstrated that the addition of TA derivatives can efficiently diminish oxygen interference. The potential NO-releasing mechanism in the presence of TA derivatives under normoxic conditions is shown in Supplementary Fig. 21b. The consumption of $^1O_2$ with the formation of superoxide anion (O₂$^{·-}$) via an ET process in the presence of TA moieties resulted in a local hypoxic microenvironment, thereby facilitating the subsequent photoredox catalysis reactions[41–43].

## Construction of micellar nanoparticles capable of releasing NO under both normoxic and hypoxic conditions

Motivated by the above results with red light-triggered NO release in organic solvents, we aimed to develop NO-releasing platforms that could be activated under both normoxic and hypoxic conditions for potential applications in highly heterogeneous biological environments, such as bacterial biofilms. We envisioned that the incorporation of NO-releasing moieties, TA moieties, and Pd-based PCs into the core of micellar nanoparticles can not only shield these reagents from the surrounding microenvironments but also promote intermolecular ET processes, enabling oxygen-independent NO release. To this end, a set of diblock copolymers (BP1-BP8) containing either NO-releasing moieties or pH-responsive TA moieties were synthesized and characterized (Supplementary Figs. 24–28) and their structural parameters are summarized in Supplementary Table 2.

To evaluate the effect of varying TA moieties on NO-releasing properties, micellar nanoparticles were co-assembled from BP1 with NO-releasing NBNO moieties and BP3-BP5 with C7A, DPA or DEA residues (1:1, wt/wt). For clarity, the resulting micellar nanoparticles were denoted as G1 (BP1/BP3), G3 (BP1/BP4), and G5 (BP1/BP5) (Table 1 and Supplementary Table 3). In addition, controlled groups of micellar nanoparticles with TA moieties but without NO-releasing moieties were also constructed as G2 (BP2/BP3), G4 (BP2/BP4), and G6 (BP2/

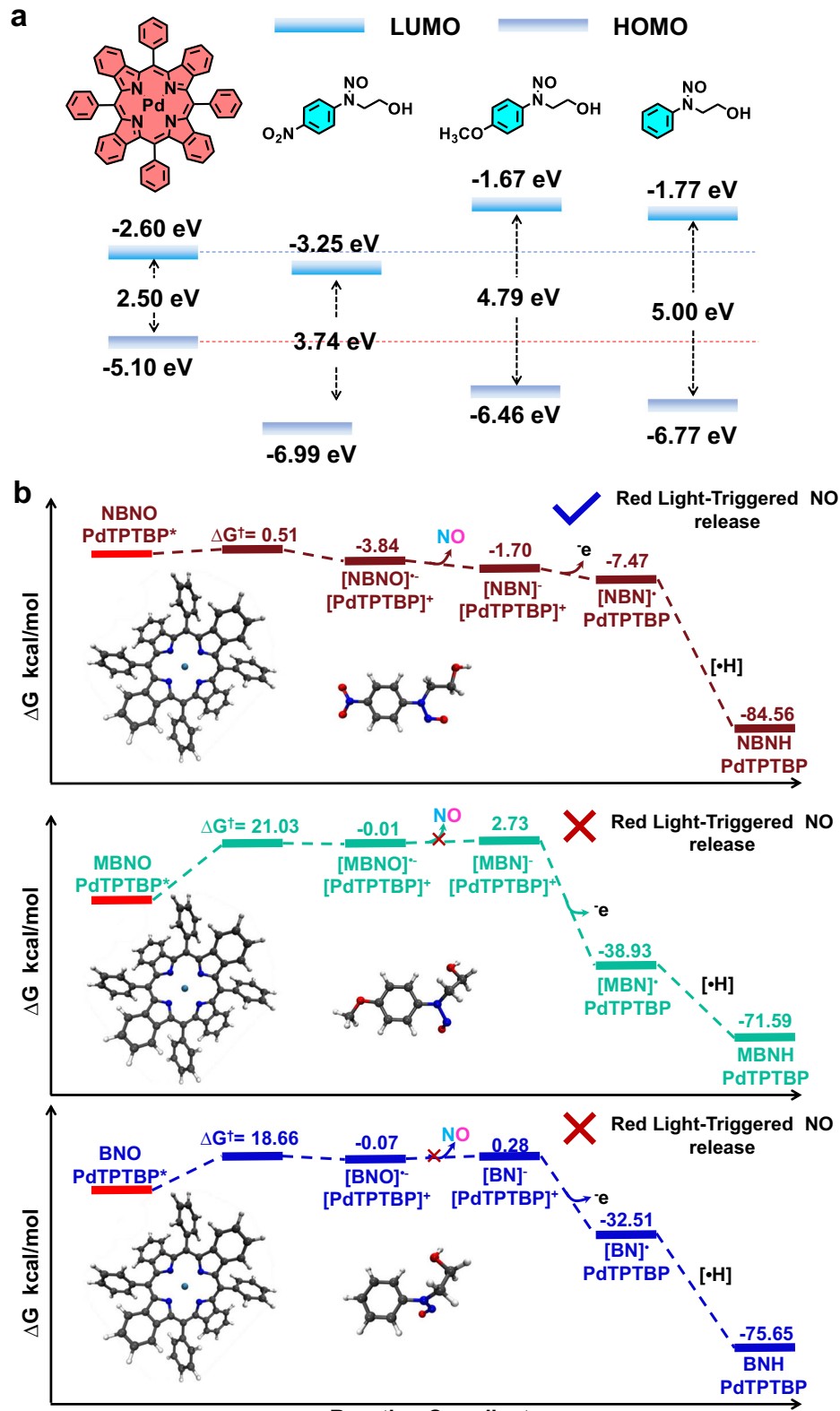

**Fig. 2 | HOMO and LUMO energy level diagram and free energy profiles. a** The relative energetic dispositions for the HOMO and LUMO of PdTPTBP, NBNO, MBNO, and BNO. **b** Free energy profiles for the PeT process from the triple excited state of [PdTPTBP]* to NBNO, MBNO, and BNO.

BP5). Dynamic light scattering (DLS) analysis demonstrated that these micellar nanoparticles exhibited hydrodynamic diameters, $<D_h>$, of ~50–100 nm, while TEM results confirmed the spherical shape of these micellar nanoparticles. Notably, these micellar nanoparticles were relatively stable in PBS buffer with no significant changes in size

observed for at least 7 days (Fig. 3a–c and Supplementary Fig. 29). Zeta-potential measurements revealed that G1, G2, G3, G4, and BP1 micelles had negative charges with zeta potentials of −14.2, −4.0, −22.6, −12.8, and −25.1 mV at pH 7.4, respectively. G5 and G6 micelles, on the other hand, displayed neutral charges with zeta potentials of 0.6 and

2.6 mV, respectively. Red light irradiation did not lead to evident changes in zeta potential values. However, upon pH decrease from 7.4 to 4.5, the zeta potentials of G1, G3, and G5 became positive (Fig. 3d and Supplementary Fig. 30b), attributed to the protonation of TA moieties within micellar nanoparticles as the p$K_a$ of BP3, BP4, and BP5 were calculated to be ~6.3, 6.0, and 7.0, respectively. Notably, the size of the micellar nanoparticles revealed negligible changes at acidic pH

### Table 1 | Summary of the micellar nanoparticles assembled from varying diblock polymers

| Entry[a] | Chemical Composition | [NBNO] (µM)[b] | [TA] (µM)[b] | [PC] (µM)[b] | [NO] (µM)[c] |
|---|---|---|---|---|---|
| G1 | BP1/BP3 | 123.2 | 242.7 | 29.6 | 85.3 (69.2%) |
| G2 | BP2/BP3 | / | 242.7 | 32.6 | / |
| G3 | BP1/BP4 | 123.2 | 265.0 | 30.7 | 27.1 (22.0%) |
| G4 | BP2/BP4 | / | 265.0 | 33.7 | / |
| G5 | BP1/BP5 | 123.2 | 269.3 | 31.0 | 48.3 (39.2%) |
| G6 | BP2/BP5 | / | 269.3 | 34.0 | / |
| / | BP1 | 246.4 | / | / | 33.6 |

[a]The block copolymers were at a weight ratio of 1/1 for G1-G6 micelles.
[b]The concentrations of NBNO, TA, and PC were calculated at a micelle concentration of 0.2 g/L.
[c]NO contents were calculated from UV-vis spectra with 630 nm light irradiation for 30 min.

values, suggesting that micellar nanoparticles did not undergo disintegration at acidic pH values (Supplementary Figs. 31, 32)[44]. It is worth noting that the acidic pH in biofilms may lead to a charge switching of micellar nanoparticles and increase the interaction between negatively charged biofilms and positively charged micelles, thereby enhancing the penetration of micellar nanoparticles in biofilms[20,45,46].

With these micelles prepared, our initial investigation focused on red light-triggered NO release in purely aqueous media under ambient conditions. It is important to highlight that BP1 micelles were unable to release NO under 630 nm light irradiation. However, the addition of a $^1O_2$-scavenging agent such as sodium ascorbate facilitated NO release, as evidenced by the UV-vis spectra. In contrast, the presence of 5,5-dimethyl-1-pyrroline n-oxide (DMPO), a superoxide anion ($O_2^{\cdot-}$) trapping agent, did not induce NO release. Therefore, the production of $^1O_2$ rather than $O_2^{\cdot-}$ under red light irradiation from PdTPTBP and the consumption of $^1O_2$ facilitated photoredox catalysis reactions and subsequent NO release under normoxic conditions (Supplementary Figs. 21 and 33).

In contrast to the requirement for additional $^1O_2$-scavenging agents, the co-assembly of BP1 and BP3 copolymers led to the formation of G1 micelles, exhibiting a marked change in UV-vis spectra under 630 nm light irradiation, in good agreement with the conversion of NBNO to NBNH moieties (Fig. 3e). The release of NO was confirmed by using a NO-specific fluorescence probe (NOFP)[47], where the formation of highly emitted deamination products was observed for G1 micelles under 630 nm light irradiation. The NO yield was 66.6% (82.1 µM)

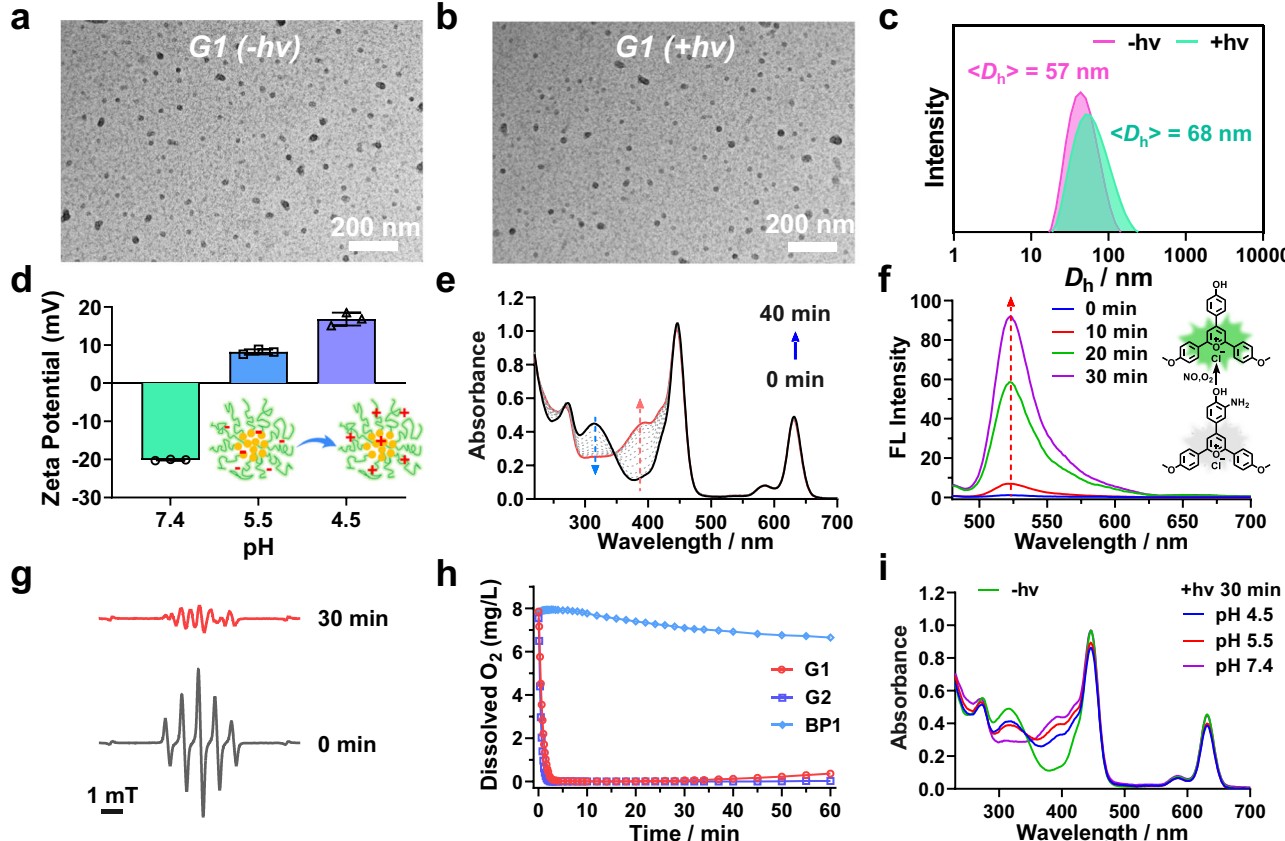

**Fig. 3 | Characterization of NO-releasing micellar nanoparticles. a, b** TEM images and (**c**) intensity-average hydrodynamic diameter distributions of G1 micelles (0.2 g/L) with or without 630 nm irradiation for 30 min. **d** ζ potentials of G1 micelles at different pH. Data are presented as mean values ± SD (*n* = 3 independent samples). **e** UV-vis spectra of G1 micelles in PBS buffer (pH 7.4) without deoxygenation under 630 nm light irradiation. **f** Evolution of fluorescence emission spectra ($\lambda_{ex}$ = 470 nm) of G1 micelles with pyrylium cation-based NO probe (NOFP) (50 µM) under 630 nm irradiation. **g** EPR spectra of G1 micelles with or without 630 nm irradiation for 30 min. **h** Changes in oxygen concentrations of aqueous dispersions of G1, G2, and BP1 micelles under 630 nm light irradiation. **i** UV-vis absorbance spectra of deoxygenated G1 micelle at varying pH values (7.4, 5.5, and 4.5) without or with 630 nm irradiation for 30 min. In all cases, the irradiation intensity was 39 mW/cm². Source data are provided as a Source Data file.

based on the standard calibration curve (Fig. 3f and Supplementary Fig. 34). Electron-spinning resonance (ESR) was employed to further validate the release of NO radicals (Fig. 3g), whereas no signal was observed for G2 and BP1 micelles (Supplementary Fig. 35), in line with the results obtained from the NBNO/PdTPTBP mixture in DMSO. These findings supported red light-triggered NO release from G1 micellar nanoparticles in the presence of TA moieties under normoxic conditions.

In addition, control experiments revealed that the UV-vis spectral changes in G3 and G5 micelles were relatively limited under the same irradiation conditions (Supplementary Fig. 36). The released NO contents were determined to be 62.6 μM (50.8%), 16.7 μM (13.6%), and 25.6 μM (20.8%) for G1, G3, and G5 micelles using the standard Griess assay (Supplementary Fig. 37). We observed a significant reduction in dissolved oxygen levels, from ~8.1 to ~0.02 mg/L within 5 min for G1 and G2 micelles under 630 nm light irradiation at pH 7.4 under open air conditions (Fig. 3h). This finding underscored the successful implementation of the photoredox catalysis process without the need for additional oxygen-scavenging agents, rendering it advantageous for potential applications in pathological tissues characterized by intricate oxygen gradients. Conversely, G3, G4, G5, and G6 micelles with DPA or DEA residues showed slower rates of oxygen depletion under identical conditions (Supplementary Fig. 38a). Notably, BP1 micelles without any TA moieties showed no significant changes in dissolved oxygen concentration levels (Fig. 3h). The formation of $^1O_2$ and subsequent formation of $O_2^{·-}$ via the reaction between $^1O_2$ and TA moieties were confirmed by the $^1O_2$-specific SOSG and EPR tests (Fig. 1, Supplementary Figs. 21, and 39–41). These results supported that incorporating TA moieties within the micelle cores enabled oxygen-tolerant photoredox catalysis in purely aqueous media, while the limited oxygen-scavenging capacities of DPA and DEA residues in G3 and G5 micelles might explain the inefficient photoredox catalysis and consequently the lower release of NO.

When the pH value was adjusted to 5.5 or 4.5 under normoxic conditions, no significant changes in the UV-vis absorbance spectra of G1 micelles were observed under 630 nm light irradiation (Supplementary Fig. 42). In addition, a decelerated rate of oxygen depletion was noted in G1 micelles (Supplementary Fig. 38b). Further analysis demonstrated that protonated C7A residues ($pK_a$ ~ 6.3) at acidic pH lost their ability to act as electron-donating agents, ultimately leading to a decreased rate of oxygen depletion[48]. Intriguingly, we noticed that the UV-vis spectra of deoxygenated micelles closely matched the absorbance spectrum of G1 micelles at pH 7.4 under normoxic conditions. This result implied that the photoredox catalysis process could also operate under acidic and hypoxic conditions (Fig. 3i), which aligned with typical pathological microenvironments such as infected and cancer tissues[11,12]. Collectively, the current NO-releasing platform can be activated by red light irradiation under both normoxic/neutral pH and hypoxic/acidic pH conditions, making it potentially suitable for heterogeneous pathological conditions.

**Red light-mediated NO release within heterogeneous biofilms**
Biofilms formed by pathogenic bacteria are implicated in a wide range of diseases. However, the significant drug resistance exhibited by biofilms greatly diminishes the bactericidal effect of antibiotics. The physiological heterogeneity within biofilms, characterized by pH and oxygen gradients, substantially reduces the antibacterial potency of antibiotics and promotes the development of antibiotic resistance[11,12,17].

To explore their antibiofilm applications, we initially examined the biocompatibility of these micellar nanoparticles. G1, G2, G3, G4, and BP1 micelles showed no noticeable hemolysis or cell toxicity when tested against both L929 and RAW264.7 cells. However, G5 and G6 micelles containing DEA residues exhibited more than 80% hemolysis and 20% cell death at a micelle concentration of 0.1 g/L

(Supplementary Figs. 43, 44). Consequently, G5 and G6 micelles were excluded from further evaluation. Considering the NO-releasing capacity of G1 micelles compared to G3 micelles (Fig. 3e and Supplementary Fig. 36), G1 micelles were selected for further investigation in antibiofilm applications.

The biofilm matrix creates a biological barrier that hinders the penetration of therapeutic agents such as antibiotics. To assess the ability of these NO-releasing micelles to penetrate mature biofilms, rhodamine B (RhB)-labeled block copolymers with similar chemical compositions (BP6 and BP8, Supplementary Figs. 25, 27) were synthesized. It is worth noting that the phosphorescence signals of PdTPTBP moieties were greatly quenched in the presence of oxygen, making them unsuitable for CLSM studies evaluating biofilm penetration. Mature CRPA biofilms were cultured and incubated with G7 (BP6/BP7), G8 (BP6/BP8), BP6, and BP8 micelles at 37 °C and the biofilm was stained with SYTO9 (green). After 1 h of incubation, the penetration depths of the G7 and G8 micelles were calculated to be ~9.8 and 8.5 μm, respectively. In sharp contrast, BP6 micelles lacking pH-responsive C7A moieties showed poor biofilm penetration (Fig. 4a, b and Supplementary Fig. 45). The increased biofilm penetration in the presence of C7A moieties can be attributed to the protonation of C7A moieties within the acidic biofilm, resulting in the formation of positive zeta potentials that increased the electrostatic interaction between micelles and biofilms (Fig. 3d)[49]. Notably, the penetration of micelles did not significantly affect the heterogeneous microenvironments within the biofilm. (Supplementary Fig. 46). Upon biofilm penetration, the hypoxic microenvironment within the inner layer of the biofilm enabled NO release even under acidic pH (Fig. 3i). Therefore, the incorporation of TA moieties served not only as an electron-donating agent to overcome oxygen quenching in photoredox catalysis under normoxic conditions but also facilitated biofilm penetration upon protonation within acidic biofilms. Under both conditions, red light-mediated NO release can be successfully achieved, which is beneficial for overcoming the heterogeneity of biofilms.

The anti-biofilm effect of G1 micelles was then evaluated. The red light-triggered NO release from G1 micelles efficiently dispersed CRPA biofilms and killed bacteria, as indicated by LIVE/DEAD® BacLight™ bacterial viability kit staining analysis (Fig. 4c and Supplementary Fig. 47). Scanning electron microscopy (SEM) analysis also revealed NO-mediated biofilm dispersal, with severe damage to CRPA morphology and loss of membrane integrity following G1 micelle treatment (Fig. 4d). Crystal violet staining revealed a 90.8% reduction in biofilm biomass after 30 min of treatment with G1 micelles under 630 nm light irradiation (Fig. 4e and Supplementary Fig. 48). Bacterial viability was further evaluated using a colony-forming-unit (CFU) assay. While G2 micelles without NO release could kill planktonic bacteria through local ROS (Supplementary Figs. 39–41), only NO-releasing G1 micelles were capable of both dispersing biofilm and killing planktonic bacteria. On the other hand, BP1 micelles without NO release under red light irradiation and treatment with Cip (25 μg/mL) failed to eradicate CRPA biofilms (Fig. 4e, f). We quantified the nitrite concentrations in the biofilms using the Griess assay and found an increased nitrite concentration only in the G1 group after 630 nm light irradiation (Fig. 4g). Furthermore, NO generation from G1 micelles within the CRPA biofilm can be easily detected through the turn-on fluorescence of the NOFP probe (Supplementary Fig. 49). Moreover, these micellar nanoparticles containing TA moieties exhibited interactions with CRPA bacteria (Supplementary Fig. 50), which may increase antibacterial activity through local NO production under red light irradiation[26]. Therefore, the current NO-releasing nanoparticles, with their adaptive properties to biofilm microenvironments, not only facilitated enhanced biofilm penetration through charge reversal but also enabled biofilm microenvironment-adaptive NO release.

To elucidate the antibiofilm mechanism of G1 micelles, the gene expression of CRPA was analyzed using RNA sequencing. Comparative

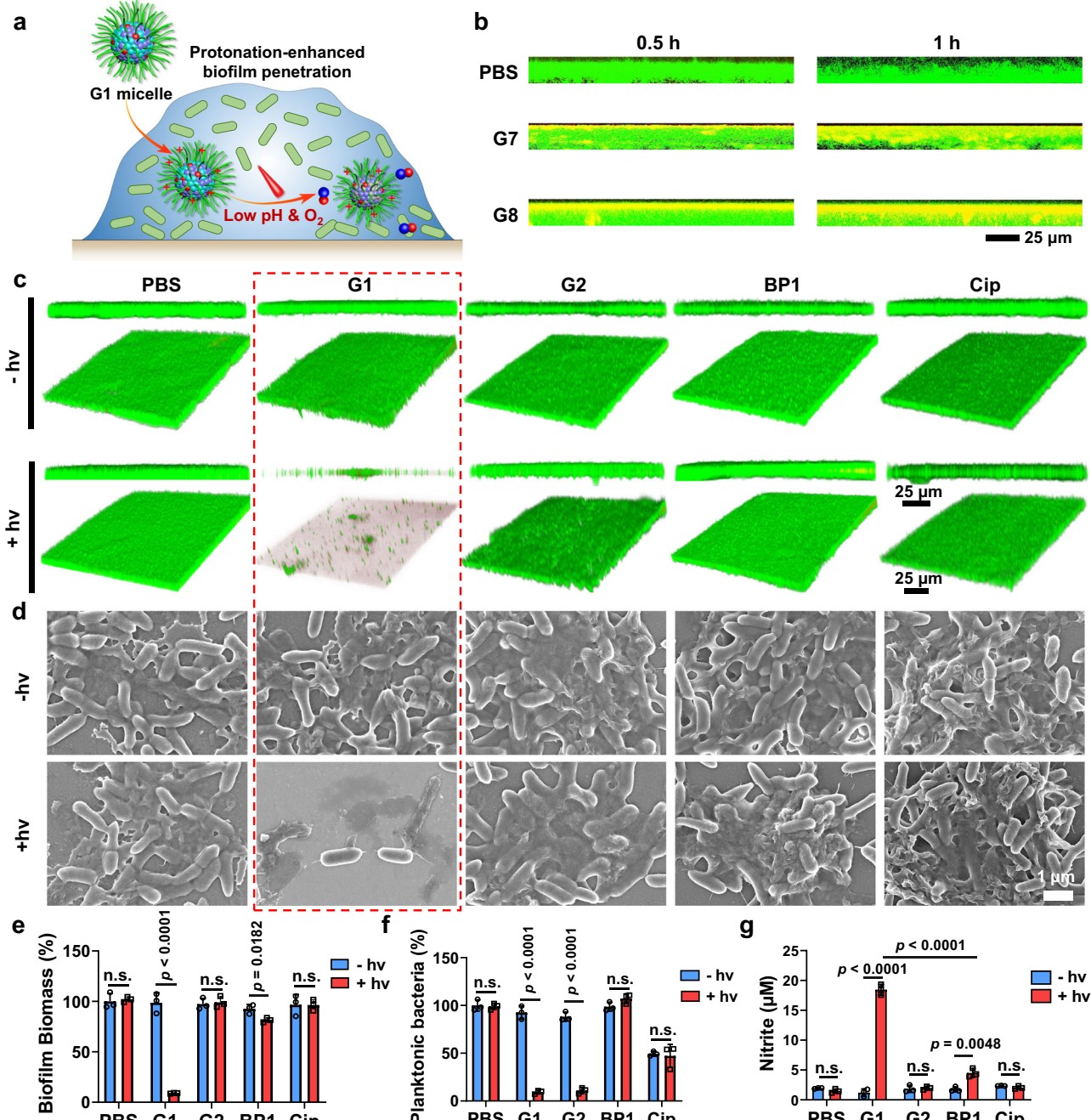

**Fig. 4 | Enhanced biofilm penetration and in vitro antibiofilm evaluation.**
**a** Schematic representation of the protonation of TA moieties in G1 micelles facilitating the penetration of micellar nanoparticles in biofilms, where the hypoxic microenvironment enabled NO release under red light irradiation. **b** Representative 3D projection of image z-stacks showing the distribution of PBS, G7, and G8 micelles (red) in CRPA biofilms (green). **c** 3D CLSM images of CRPA biofilms stained with the LIVE/DEAD® BacLight™ bacterial viability kit and (**d**) SEM images of CRPA biofilms after treatment with G1, G2, BP1 micelles and free Cip with or without 630 nm light irradiation for 30 min. **e** Bacterial biomass of CRPA biofilms quantified by crystal violet staining, (**f**) bacterial viability of planktonic bacteria, and (**g**) nitrite concentrations in biofilm suspensions by Griess assay after treatment with G1, G2, BP1 micelles and free Cip with or without 630 nm light irradiation. Data are presented as the mean values ± SD ($n = 3$ independent samples). n.s., not significant. Statistical analysis was calculated by two-tailed Student's $t$-test. Source data are provided as a Source Data file.

analysis between the groups treated with G1 micelles and 630 nm light irradiation versus PBS revealed that 266 genes were downregulated, while 354 genes were upregulated (Fig. 5a). Notably, the NO release from G1 micelles led to significant downregulation of genes associated with multidrug resistance (e.g., *mexC* and *mexD*), and the expression of these genes is closely associated with multidrug resistance efflux pumps, resulting in a significant reduction in CRPA drug resistance (Fig. 5b and Supplementary Fig. 51)[14]. In addition, the expression of

genes related to biofilm formation (e.g., *dgcA*) was largely downregulated (Fig. 5b, c)[50–52]. It is noteworthy that the *dgcA* gene mediates the synthesis of DGC protein, which plays a crucial role in the synthesis of cyclic di-GMP (c-di-GMP). While c-di-GMP serves as a near-ubiquitous second messenger that governs a multitude of bacterial behaviors and holds pivotal significance in orchestrating the transition between motile loner cells and biofilm formers[53], the reduction in c-di-GMP levels eventually induces the dispersion of biofilms[54,55]. Moreover,

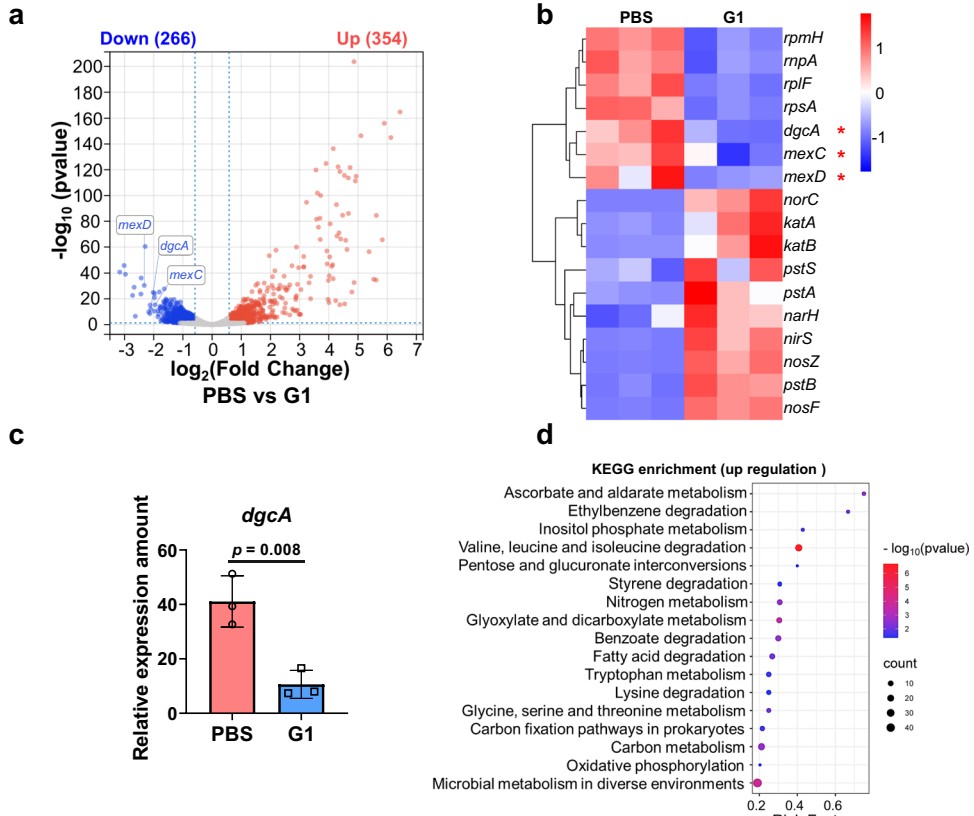

**Fig. 5 | Gene expression in CRPA bacteria. a** Volcano plots of differentially expressed genes (gray: not significantly different genes; red: upregulated genes; blue: downregulated genes), (**b**) heatmap of differentially expressed genes, (**c**) relative gene expression of *dgcA* and (**d**) KEGG enrichment of upregulated genes of CRPA treated with PBS or G1 micelles with 630 nm light irradiation for 30 min (39 mW/cm²). Data are presented as the mean values ± SD (*n* = 3 independent samples). Statistical analysis was calculated by two-tailed Student's *t*-test. Source data are provided as a Source Data file.

Kyoto Encyclopedia of Gene and Genomes (KEGG) signaling pathway analysis indicated that pathways related to carbon metabolism and the tricarboxylic acid cycle exhibited significant upregulation and enrichment (Fig. 5d). These findings supported that NO release from G1 micelles effectively eradicated biofilms and inhibited the development of antibiotic-resistant bacteria.

## Antibacterial evaluation in a CRPA biofilm-infected mouse model

Encouraged by the excellent antibiofilm and bactericidal performance observed in vitro by using biofilm microenvironment-adaptive NO release, we further assessed the anti-infectious capacity of G1 micelles in vivo against CRPA biofilm infections in an open skin wound model[56]. Notably, in an animal wound model with open wound biofilm infections, the wound area's complex microenvironment often presents challenges for biofilm infection-associated wound treatment (i.e., neutral pH/normoxia outside the biofilm and acidic pH/hypoxia within the biofilm)[57]. Previous evidence suggested that biofilm formation on open wounds for *P. aeruginosa* occurred after 8 h of inoculation[58]. We extended the inoculation period to 12 h before applying any treatment. After establishing a full-thickness skin wound infection mouse model (~6 mm in diameter), the mice were treated with G1, G2, and BP1 micelles with or without light irradiation, and Cip (25 µg/mL). G1 micelles with 630 nm light irradiation exhibited antibacterial activity and accelerated wound closure compared to Cip, while G1 micelles without irradiation and G2 and BP1 micelles with or without 630 nm light irradiation showed similar results to the PBS group (Fig. 6a–d and Supplementary Figs. 52, 53). This finding demonstrated that G1 micelles overcame the heterogeneous oxygen gradients and enabled local NO release in a drug-resistant biofilm infected model.

Importantly, there were no substantial changes in body weight during the entire treatment process, indicating good biocompatibility of NO-releasing micelles (Fig. 6e and Supplementary Fig. 52d). Histological and immunohistological fluorescence analyses revealed that red light-triggered NO release from G1 micelles enhanced collagen deposition (Masson staining), promoted neovascularization (CD31 staining), and reduced inflammatory factors such as tumor necrosis factor-alpha (TNF-α staining) in wound tissues (Fig. 6f and Supplementary Figs. 54, 55). Therefore, NO-releasing micelles capable of adapting to heterogeneous biofilm microenvironments present a promising strategy to combat biofilm infections with minimal toxicity.

## Discussion

In summary, we successfully developed biofilm microenvironment-adaptive micellar nanoparticles capable of releasing NO through two distinct photoredox catalysis mechanisms. Importantly, the incorporation of TA moieties within these micellar nanoparticles played a critical role as electron-donating agents, enabling the circumvention of oxygen quenching of PCs under neutral and normoxic conditions. Additionally, these TA moieties served as proton acceptors, facilitating deep biofilm penetration upon protonation without compromising photoredox catalysis reactions within acidic and hypoxic biofilms. Consequently, the bespoke micelles achieved oxygen and pH-independent NO release to overcome biofilm heterogeneity, showing excellent antibiofilm and bactericidal effects against CRPA infections without the need for conventional antibiotics. Given the physiological heterogeneity observed in many diseases, this NO-releasing platform with adaptive properties to pathological microenvironments holds great promise not only for combating biofilm infections but also for treating cancer-related diseases.

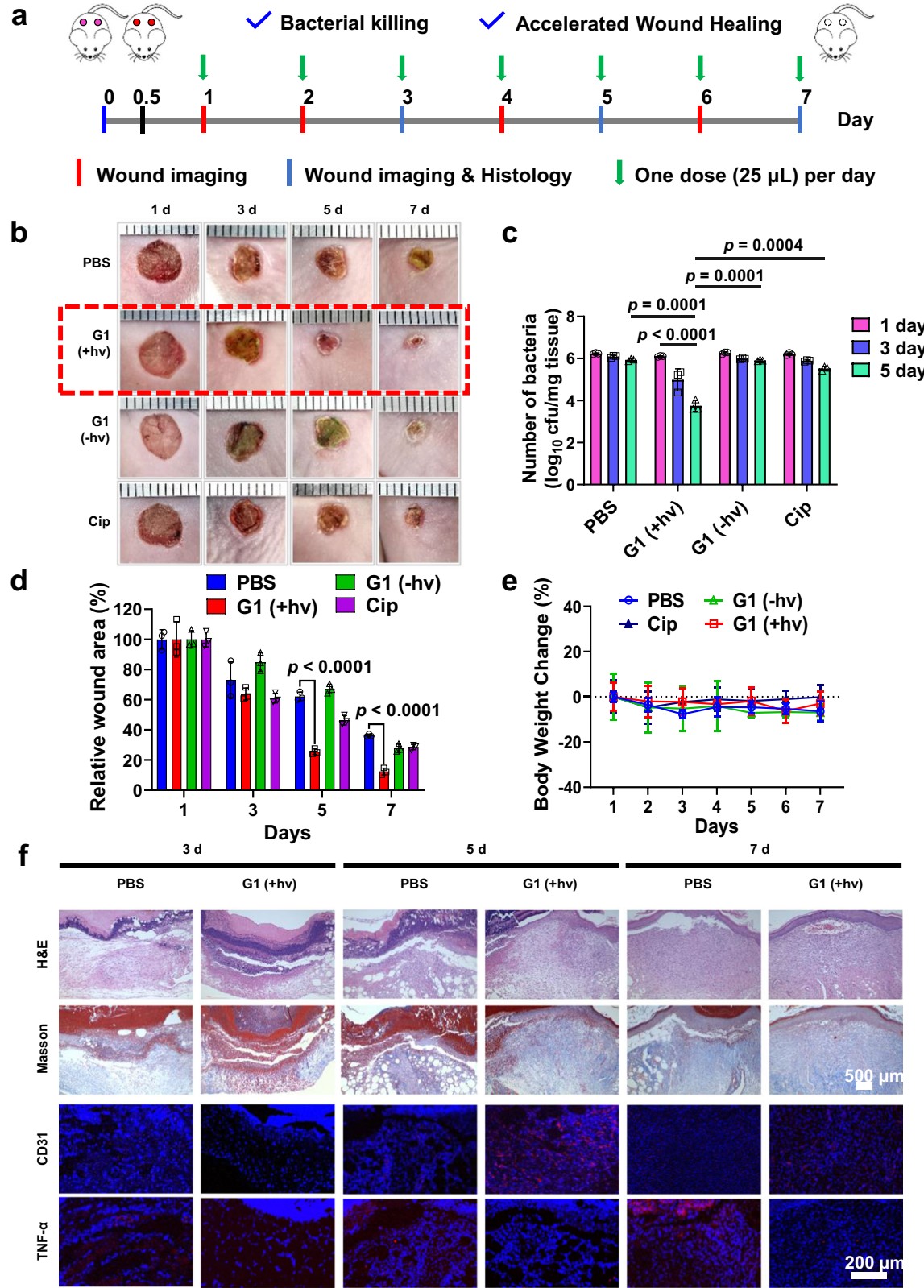

**Fig. 6 | In vivo anti-biofilm evaluation in CRPA-infected mice. a** Experimental outline of in vivo antibacterial assessment in a CRPA-infected wound healing model. **b** Representative skin wound images and (**c**) bacterial colony-forming units separated from wound tissues, (**d**) residual wounded areas, and (**e**) changes in body weights of CRPA-infected mice after treatments with PBS, and G1 micelles without (G1 - hv) and with (G1 + hv) 630 nm light irradiation for 30 min. **f** Histological and immunofluorescence analysis of the CRPA-infected mice receiving PBS or G1 + hv treatments. Data are presented as the mean values ± SD ($n = 3$ independent samples). Statistical analysis was calculated by two-tailed Student's *t*-test. Source data are provided as a Source Data file.

## Methods

### Fabrication of G1 micelles through the co-assembly of BP1 and BP3

BP1 (1 mg) and BP3 (1 mg) were dissolved in THF (1 mL). The resulting solution was added dropwise to MilliQ water under sonication. The resulting mixture was then transferred into a dialysis tube (MWCO = 14 kDa). The dialysis process was carried out against DI water for a duration of 12 h to remove the organic solvent. Fresh DI water was replenished approximately every 3 h during the dialysis process. This procedure resulted in the formation of G1 micelles. The fabrication of G2, G3, G4, G5, G6, G7, and G8 micelles followed the above procedures using corresponding copolymers.

### General procedures for bacterial biofilm formation and harvesting

CRPA was used for biofilm experiments. In each well of a 24-well plate, LB medium (400 μL) and CRPA suspension ($1 \times 10^8$ cfu/mL, 100 μL) were added. The plate was placed in an incubator and cultured at 37 °C for 24 h. Subsequently, the aged medium in each well was carefully removed and replaced with fresh LB medium. The biofilms were allowed to continue growing by culturing for an additional 12 h. The resulting biofilms, obtained after 12 h of incubation, were then used for further antibiofilm experiments.

### Biofilm penetration of micellar nanoparticles

To investigate the interaction between micelles and CRPA biofilms, CRPA biofilms were cultured according to the above method. The biofilms were exposed to PBS buffer containing G7, G8, BP6, and BP8 micelles for 30 and 60 min. After exposure, the biofilms were stained with SYTO9 according to the manufacturer's instructions. CLSM was used to study the penetration of micelles into the biofilms. The SYTO9 channel was excited at 488 nm and collected at 500-550 nm. The RhB channel was excited at 543 nm and collected at 560−590 nm.

### Quantification of biofilm biomass by crystal violet staining

The cultured CRPA biofilms were treated with G1, G2, and BP1 micelles, and free Cip (25 μg/mL) for 1 h at 37 °C. After various treatments, the biofilms were subjected to 630 nm light irradiation (39 mW/cm$^2$) for 30 min, while the non-irradiated biofilms were used as the control. Subsequently, the biofilms were incubated at 37 °C for 60 min, and planktonic bacteria were removed. The biofilms were then washed with PBS. To assess the biofilm mass amounts, crystal violet staining was performed. First, 0.5 mL of a crystal violet staining agent (0.1 wt% in PBS) was added to each well containing the biofilms. The plate was incubated for 20 min to allow staining of the biofilms. Afterward, the wells were washed with PBS twice to remove the excess crystal violet staining agent. The crystal violet was then dissolved in pure ethanol (500 μL), and the optical density at 550 nm (OD550) of this solution was measured using a microtiter plate reader (Thermo Fisher).

### Evaluation of bacterial viability by colony forming unit assay

To evaluate the bacterial viability in the CRPA biofilms, the standard colony-forming unit (CFU) assay was performed. Following various treatments, the planktonic bacteria present in the culture medium were diluted and plated on Trypticase soy broth (TSB) agar plates. These plates were then incubated at 37 °C for 18 h to facilitate the growth of bacterial colonies. The colonies on the plates were counted to determine the number of viable bacteria in the planktonic fraction. For the biofilm fraction, the biofilms were initially washed three times with PBS to remove any non-adherent bacteria. The biofilms were subjected to ultrasound treatment (200 W, 40 kHz) in 0.5 mL PBS for 30 min. After the ultrasound treatment, the cell suspension containing the detached bacteria was serially diluted and plated onto TSB agar plates. The plates were then incubated at 37 °C for 18 h to allow the bacterial colonies to grow. The colonies on the plates were counted to determine the number of viable bacteria in the biofilm fraction.

### Live/dead staining CRPA biofilms

The CRPA biofilms were cultured according to the above method. After applying various treatments to the biofilms, they were washed three times with PBS to remove any residual micelles that might be present on the surface of the biofilms. The biofilms were stained using LIVE/DEAD® BacLight™ bacterial viability kit reagents (Molecular Probes) following the instructions provided by the manufacturer. The biofilms were incubated at room temperature for 20 min under dark conditions. After staining, the biofilms were observed using CLSM. SYTO9 was excited using a 488 nm laser, and the emitted fluorescence was collected at 500-550 nm. Propidium iodide (PI) was excited using a 543 nm laser, and the emitted fluorescence was collected at 600−650 nm.

### Variations of bacterial morphologies in biofilms

The CRPA biofilms were grown on sterile cover slides in 24-well plates according to the above method. After the pre-determined treatments, the biofilm samples were washed with PBS, and then immediately fixed with 2.5% paraformaldehyde at 4 °C overnight. After fixation, the samples were washed with water and then with a series of graded ethanol solutions (30%, 50%, 70%, 80%, 90%, 95%, and 100%) and dried for SEM observation.

### RNA sequence

For RNA-seq analysis, the entire RNA was extracted using the TRIzol RNA Extraction Kit. All gene profiles were detected by GENEWIZ Co., Ltd. (Suzhou, China). Differential expression analysis used the DESeq2 Bioconductor package, a model based on the negative binomial distribution. To control for false positives in the analysis, the p-values of the differentially expressed genes were adjusted using the Benjamini and Hochberg approach. In this analysis, genes with an adjusted p-value (Padj) < 0.05 were considered significantly differentially expressed. The GOSeq package (v1.34.1) was used to identify Gene Ontology (GO) terms that annotate a list of enriched genes with a significant p-value less than 0.05.

### In vivo antibacterial study

All the animal studies described in this research were approved by the Committee on the Ethics of Animal Experiments of the University of Science and Technology of China (USTC) and were performed in strict accordance with the Animal Care and Use Committee of USTC. All animals were maintained on a 12-12 light-dark cycle with a temperature of 25 °C and humidity of 48−52%. Dorsal hair BALB/c mice (6–8 weeks, purchased from the Experimental Animal Center of Anhui Medical University) were shaved, and full-thickness circular wound were created on the back of each mice using surgical scissors (6 mm in diameter, $n = 4$)[59]. The operation was performed under aseptic conditions before the infection. After wound formation, 20 μL of CRPA suspensions containing $10^7$ cfu/mL in PBS was deposited onto the wounds. After 12 h of inoculation, wounds were treated with 25 μL of G1 micelles with or without 630 nm irradiation (39 mW/cm$^2$) for 30 min, G2 micelles with or without 630 nm irradiation (39 mW/cm$^2$) for 30 min, BP1 micelles with or without 630 nm irradiation (39 mW/cm$^2$) for 30 min, Cip (25 μg/mL, MBC) and PBS. On days 1, 3, and 5, the wound tissues were excised, weighed, and homogenized in PBS. The homogenate was plated on TSB agar for cfu checking. On days 3, 5, and 7, the wound and surrounding tissues were harvested and fixed with 4% paraformaldehyde overnight, embedded in paraffin, and sectioned for further histological evaluation.

**Computational details**

All density-functional theory (DFT) calculations were performed at the level of B3LYP/6-31 G + (d, p) (LANL2DZ on Pd) in IEFPCM-DMSO solvent using the Gaussian 09 software package [Frisch, M. J.; Trucks, G. W.; Schlegel, H. B.; Scuseria, G. E.; Robb, M. A.; Cheeseman, J. R.; Scalmani, G.; Barone, V.; Mennucci, B.; Petersson, G. A.; Nakatsuji, H.; Caricato, M. Gaussian 09, d.01; Gaussian, Inc.: Wallingford, CT, 2013]. We initially optimized the ground state geometry for the PdTPTBP photosensitizer and NBNO, MBNO, and BNO, and the optimized ground state structures were then submitted for time-dependent density-functional theory (TD-DFT) calculations to derive their energy levels of the S1 and T1 states.

**Statistical analysis**

Data are presented as the mean ± standard deviation from at least three parallel experiments. Statistical analysis was evaluated by 8.0 software (GraphPad, San Diego, California, USA) and Student's $t$-test. $p < 0.05$ was considered statistically significant.

**Reporting summary**

Further information on research design is available in the Nature Portfolio Reporting Summary linked to this article.

## Data availability

The RNA-Seq data generated in this study have been deposited in the NCBI Gene Expression Omnibus database under the accession code GSE246965. The experimental data generated in this study are available within the Article and Supplementary Information. All other data are available from the corresponding authors upon request. Source data are provided with this paper.

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

## Acknowledgements

This work was supported by the National key R&D Program of China (2020YFA0710700 to S.Y.L.), the National Natural Scientific Foundation of China (52233009 to S.Y.L., 52021002 to S.Y.L., U19A2094 to S.Y.L., 52273155 to J.M.H., 52073270 to J.M.H., and 51973206 to G.Y.Z.), the Strategic Priority Research Program of the Chinese Academy of Sciences (XDB0450301 to S.Y.L. and XDB0450102 to J.M.H.), the Joint Funds from Hefei National Synchrotron Radiation Laboratory (KY2060000197 to J.M.H.), the Collaborative Innovation Program of Hefei Science Center, CAS (2022HSC-CIP012 to J.M.H.), and Fundamental Research Funds for the Central Universities (YD9110002025 to J.C.).

## Author contributions

J.H., S.L., and C.Z. supervised the project. J.H. and J.C. conceived the experiments; J.C., G.G., and S.Z. carried out the experiments; J.H., G.Z. help with the data interpretations. J.H., and S.L. wrote the manuscript. All authors discussed the results and commented on the paper.

## Competing interests

The authors declare no competing interests.
