## [Peer Review File · Nature Communications]

Biofilm Heterogeneity-Adaptive Photoredox Catalysis Enables Red Light-Triggered Nitric Oxide Release for Combating Drug-Resistant InfectionsREVIEWER COMMENTS

Reviewer #1 (Remarks to the Author):

In this manuscript, the authors described that the synthesis of the nanoparticle which consisted of block copolymers bearing Pd-photocatalyst moieties, NBNO moieties (NO-releasing moieties), and tertiary amine moieties. They also evaluate the NO releasing ability from those nanoparticles, and biofilm penetration of the nanoparticles, and the suppression of biofilm and bacteria colonies via photoinduced NO release from the nanoparticles with red light irradiation. The experimental procedures are technically sound although some experimental methods should be appropriately added, and the results are shown in the appropriate ways. The photo-responsive NO releasing nanoparticle showed nice NO releasing ability, biofilm penetration activity, and the degradation activity of biofilm and suppression of bacterial colonies. The authors examined and elucidated that those NO releasing and antibacterial effects are largely depending on the coexistence of tertiary amine moieties in the nanoparticles. This study would be important for the multidrug resistance issues and attractive to the researcher in those field. However, the basic strategy of the nanoparticle construction is the combination of the known technology such as N-nitrosoaryl moiety for redox dependent NO release via photoredox catalyst and the suppression of oxygen-dependent photocatalyst scavenging with tertiary amine additives. The work would make a nice contribution to the field of biofilm degradation but to the limited extent. The detailed comments are listed below.

Comments,

1) As for Fig. 1 and the last paragraph of the introduction section, the authors mentioned the tertiary amine worked as a scavenger for singlet oxygen produced from photocatalytic reduction of molecular oxygen. The authors explained that this scavenging effect might facilitate photocatalytic nitric oxide production. However, in the open system like biological milieu, the scavenging process does not directly inhibit the photocatalytic energy transfer of molecular oxygen. Local hypoxic environment would be highly depending on the reaction rate of tertiary amine with singlet oxygen as well as molecular oxygen with Pd catalyst. The authors are requested to discuss the rationale of the effect of tertiary amine further in detail.

In addition, the proximity production of amine cation radical, which is a primary product of ROS scavenging, may react with nitric oxide via radical-radical interaction in nanoparticles. It would be also expected to be discussed.

2) As for Ref. 35, the authors referred to this paper for explaining N-nitrosophenyl moiety, but the paper would not directly showing the NO release from N-nitrosophenyl moiety. The authors should check the reference to inform the readership appropriate papers.

3) As for the mechanism of NBNO reduction by PdTPTBP, the authors proposed the secondary redox reaction between [NBN]⁻ and [PdTPTBP]^{+•} to afford [NBN][•]. However, the primary product, [NBN]⁻ would be easily and rapidly protonated under aqueous conditions, especially in acidic conditions, and then [PdTPTBP]^{+•} may oxidize other components including biological ones around the PC. The authors should carefully consider the possible reactions in their reaction system.

- 4) As for the amount of NO release from G1, G3, and G5, the authors calculated the NO amount from the absorption change of NBNO and NBNH in accordance with the method shown in Supplementary information. The amount of effective NO concentration would not be appropriately estimated from such production formation. The authors should measure the amount of NO release by direct measurement with NO electrode, or by indirect methods such as ESR spin trapping method, fluorescence probes, or nitrite/nitrate formation, as the authors performed them in bacterial experiments.
- 5) As shown in Supplementary Figs. 31-32, the authors mentioned the integration of the nanoparticles did not so changed under acidic conditions. This result would mean the positive charges depending on the protonation of TA moiety might occur only at the surface of the nanoparticle. The deprotonated form of TA moiety inside the nanoparticle may affect the liberation of NO from nanoparticles such as oxidation of TA by NO or so. The authors should consider and discuss the possibility of the interference of NO release from the nanoparticles under both neutral and acidic conditions.
- 6) As for Fig. 3f, the authors should refer to the appropriate paper for NOFP, and also clarify the quantitative results of NO formation via NOFP, which would reflect the NO releasing efficiency in the biological conditions.
- 7) The authors should mention the methods for NO measurement such as ESP spin trapping measurement as well as NOFP.
- 8) As for Fig. 3h, the authors should clarify and measurement methods for the oxygen concentration measurement.
- 9) In accordance with Supplementary Figs. 31-32, the nanoparticles are still assembled under acidic conditions, which would mean that the TA moiety at the surface of the nanoparticle is only protonated. However, in Supplementary Fig. 37, the oxygen consumption was found to be decreased under acidic conditions at a significant extent, which may mean that a significant population of TA was protonated. This result would be not fully consistent with those suggested from Supplementary Figs. 31-32. The authors should explain and comment on these points.
- 10) In general, the nanoparticles in this study seem larger in size than known antibacterial agents. The authors should explain whether these nanoparticles can penetrate biofilms practically even with such large size, or only attached on the surface of biofilms and degraded them from the outer-rim of biofilms.
- 11) In accordance with Fig. 4b and Supplementary Fig. 44, the nanoparticles looked to access the artificial biofilm from its surface. Is the biofilm assembly affected with the penetration of the nanoparticles? If so, the oxygen concentration and pH values may almost same as that outside of the biofilm. The authors are expected to consider such possibility.
- 12) As for the results in Fig. 5, the authors should refer to the appropriate papers for the biological effects of NO on the expression of the genes of drug resistance and biofilm formation. NO may affect the biofilms via chemical degradation or via inhibition of enzymes, in which the gene expression may rather increase through bacterial response to NO stimuli. The authors should consider and explain those possibilities.

Reviewer #2 (Remarks to the Author):

This manuscript describes a micelle nanoparticle adapted to the heterogeneous biofilm microenvironment of drug-resistant bacteria that releases nitric oxide against the biofilm through two different photoredox catalytic mechanisms. The new material has potential implications for treating drug-resistant bacteria. In general, the manuscript content is very complete and comprehensive. Therefore, I propose to publish it with minor modifications. Here are some concerns:

- 1: Table 1 provides the release of nitric oxide from G1 to G6. I suggest the authors show the release curve of nitric oxide, including micelles (G1-G6) and diblock copolymers (BP1-BP8).
- 2: In Figure 4c, the fluorescence shown by the staining of the LIVE/DEAD bacterial viability kit is only one kind of fluorescence. Please provide the staining of live bacteria and the fluorescence of dead bacteria.
- 3: It can be seen from Figure 4 that G1 has a good effect against biofilm, but the figure cannot reflect the dissipating biofilm effect of nitric oxide, whether it can provide the biofilm thickness after the materials act on the biofilm.
- 4: The histological and immunofluorescence analysis in Fig. 6f and Supplementary Fig. 52 Show the intensity of CD31 and TNF- α . Please explain why the intensity of CD31 rises first and then decreases but the intensity of TNF- α tends in the opposite trend.
- 5: The material's view on heterogeneous biofilms is novel, but in the anti-biofilms and animal parts, please discuss the advantages of the material at different pH and oxygen concentrations.

Reviewer #3 (Remarks to the Author):

The noteworthy results are that NO release can be initiated from an N-nitrosoamine in the presence of red light with the goal to treat biofilm infections. The difference between this work and their previous work, published in JACS, is that in these studies, nanoparticles were produced by co-assembling diblock polymers that contained the key delivery and release components: Pd-photocatalyst, tertiary amines and an N-nitrosoamine. The nanoparticles were charged to encourage penetration into the biofilm for optimal place for biofilm dispersion. While in the films, the light caused the activation of the photocatalyst which then led to NO production. The NO production levels were high enough to disperse the biofilm. The evidence for NO release is the decrease in the UV-signal of the functional group. Finally, the manuscript describes the effectiveness of these nanoparticles on biofilm eradication in an in vivo model. This is a very complete study from design through application. The measurements were all done in at least triplicate with the average and standard deviation reported. The results are confirmed by the evidence (appropriate characterization tools used). If there was one suggestion it would be to see if they could measure the NO itself in vitro through a selective sensor or chemiluminescence based method.

While UV vis analysis of the product formed after analysis is of high value, the biouseful amount of NO may also provide some insights into the utility of these materials for other applications as well. Indeed, this is a thoughtful and well designed study.

Our replies to the reviewers' comments are below.

Reviewer #1:

In this manuscript, the authors described that the synthesis of the nanoparticle which consisted of block copolymers bearing Pd-photocatalyst moieties, NBNO moieties (NO-releasing moieties), and tertiary amine moieties. They also evaluate the NO releasing ability from those nanoparticles, and biofilm penetration of the nanoparticles, and the suppression of biofilm and bacteria colonies via photoinduced NO release from the nanoparticles with red light irradiation. The experimental procedures are technically sound although some experimental methods should be appropriately added, and the results are shown in the appropriate ways. The photo-responsive NO releasing nanoparticle showed nice NO releasing ability, biofilm penetration activity, and the degradation activity of biofilm and suppression of bacterial colonies. The authors examined and elucidated that those NO releasing and antibacterial effects are largely depending on the coexistence of tertiary amine moieties in the nanoparticles. This study would be important for the multidrug resistance issues and attractive to the researcher in those field. However, the basic strategy of the nanoparticle construction is the combination of the known technology such as N-nitrosoaryl moiety for redox dependent NO release via photoredox catalyst and the suppression of oxygen-dependent photocatalyst scavenging with tertiary amine additives. The work would make a nice contribution to the field of biofilm degradation but to the limited extent. The detailed comments are listed below.

Reply: We express our profound gratitude for your valuable comments on the manuscript. With your guidance, we have revised the pertinent sections of the manuscript, meticulously addressing each of your inquiries individually.

1. As for Fig. 1 and the last paragraph of the introduction section, the authors mentioned the tertiary amine worked as a scavenger for singlet oxygen produced from photocatalytic reduction of molecular oxygen. The authors explained that this scavenging effect might facilitate photocatalytic nitric oxide production. However, in the open system like biological milieu, the scavenging process does not directly inhibit the photocatalytic energy transfer of molecular oxygen. Local hypoxic environment would be highly depending on the reaction rate of tertiary amine with singlet oxygen as well as molecular oxygen with Pd catalyst. The authors are requested to discuss the rationale of the effect of tertiary amine further in detail. In addition, the proximity production of amine cation radical, which is a primary product of ROS scavenging, may react with nitric oxide via radical-radical interaction in nanoparticles. It would be also expected to be discussed.

Reply: (a) We appreciate these thoughtful comments. The reviewer is definitely right on this point and we agree with the reviewer's opinion that the scavenging of singlet oxygen ($^1\text{O}_2$) by tertiary amines (TA) does not directly inhibit the photocatalytic energy transfer to molecular oxygen. Upon light activation, the photosensitizer PdTPTBP can undergo type II reactions to produce $^1\text{O}_2$. The electron-rich TA moieties functions as an electron donor and can scavenge $^1\text{O}_2$, resulting in the generation of superoxide anion ($\text{O}_2^{\cdot-}$) and eventually hydrogen peroxide (H_2O_2) via the dismutation reactions of $\text{O}_2^{\cdot-}$ (Ferji et al., *Macromolecules* **2019**, 52, 6898; Kwon and Boyer et al. *Chem. Soc. Rev.* **2023**, 52, 3035). Indeed, it is noteworthy that TA acts as a scavenger for $^1\text{O}_2$ rather than $^3\text{O}_2$, the oxidation of amines by molecule oxygen ($^3\text{O}_2$) at room temperature is indeed slow (Yoon et al., *Science* **2014**, 343, 1239176). Therefore, as

proposed by the reviewer, the presence of TA does not directly inhibit the energy transfer from the photocatalyst to $^3\text{O}_2$ but scavenges the locally formed $^1\text{O}_2$ to produce a hypoxic microenvironment, facilitating the subsequent photoredox catalysis between the photocatalyst and NO donor moieties, resulting in photo-mediated NO release. To further support this claim, we used another $^1\text{O}_2$ -scavenging agent, sodium ascorbate (SA), and a similar change in oxygen concentrations to **G1** micelles under 630 nm light irradiation was observed.

Supplementary Fig. 38. (a) Changes in dissolved oxygen of aqueous dispersions of **G3-G6** micelles (0.2 g/L) in PBS (pH 7.4, 10 mM) under 630 nm light irradiation. (b) Changes in dissolved oxygen of aqueous dispersions of **G1** micelles (0.2 g/L) at pH 7.5, 5.5, 4.5, and **BP1** micelles in the presence or absence of sodium ascorbate (10 mM) under 630 nm light irradiation (39 mW/cm²).

(b) With regard to the possible reaction between $[\text{TA}]^+$ and NO radical, this reaction was thermodynamically unfavorable according to DFT calculations ($\Delta G_{\text{III}} = 3.57 \text{ kcal/mol} > 0$; Supplementary Fig. 21b). Therefore, this process is less likely to happen under the current situation.

Supplementary Fig. 21. Proposed mechanisms of the activation of NBNO with PdTPTBP in (a) hypoxic and (b) normoxic conditions. Inset: detection of H₂O₂ using a test strip (D, dark; L, light).

- As for Ref. 35, the authors referred to this paper for explaining N-nitrosophenyl moiety, but the paper would not directly showing the NO release from N-nitrosophenyl moiety. The authors should check the reference to inform the readership appropriate papers.

Reply: Thanks for pointing this oversight out. We have updated the original ref 35 (Miyata et al., *J. Am. Chem. Soc.* **2005**, *127*, 11720) with the following reference (Nakagawa et al., *J. Am. Chem. Soc.* **2014**, *136*, 7085; now as ref. 34 in the revised version).

3. As for the mechanism of NBNO reduction by PdTPTBP, the authors proposed the secondary redox reaction between [NBN]⁻ and [PdTPTBP]^{+•} to afford [NBN][•]. However, the primary product, [NBN]⁻ would be easily and rapidly protonated under aqueous conditions, especially in acidic conditions, and then [PdTPTBP]^{+•} may oxidize other components including biological ones around the PC. The authors should carefully consider the possible reactions in their reaction system.

Reply: We appreciate the reviewer's thoughtful comments. We fully agree with the reviewer's opinion that [NBN]⁻ could undergo rapid protonation and [PdTPTBP]^{+•} could be likely oxidized by certain oxidative agents under complex biological microenvironments. When the photoredox catalysis reaction was implemented in PBS buffer at neutral pH and devoid of biomolecules, we hypothesized that [NBN]⁻ could mostly likely be oxidized by [PdTPTBP]^{+•}. To make the proposed reaction mechanism concise, we did not incorporate the potential reactions of these intermediates with biological molecules. However, we have followed the reviewer's suggestion to include the above discussion in the revised manuscript to make it more accurate.

"It is worth noting that more complex reaction mechanisms could be involved under truly biological conditions. For example, [NBN]⁻ would be readily protonated under aqueous conditions, especially in pathological acidic conditions, and [PdTPTBP]^{+•} could likely be oxidized by other biological oxidative agents as well. (page 9, 1st paragraph)"

4. As for the amount of NO release from G1, G3, and G5, the authors calculated the NO amount from the absorption change of NBNO and NBNH in accordance with the method shown in Supplementary information. The amount of effective NO concentration would not be appropriately estimated from such production formation. The authors should measure the amount of NO release by direct measurement with NO electrode, or by indirect methods such as ESR spin trapping method, fluorescence probes, or nitrite/nitrate formation, as the authors performed them in bacterial experiments.

Reply: Many thanks for these valuable suggestions. According to the reviewer's insightful suggestions, we employed the standard Griess assay to measure the NO release contents of these micelles under 630 nm light irradiation (**Supplementary Fig. 37**). Specifically, the NO release efficiency was determined to be 62.6 μM (50.8%), 16.7 μM (13.6%), and 25.6 μM (20.8%) for **G1**, **G3**, and **G5** with 30 min of light irradiation, respectively. It should be mentioned that the NO-releasing efficiencies calculated by the Griess assay were lower than those determined by UV-vis spectrometry. This result was reasonable because the Griess assay can only estimate the nitrite concentration, leading to underestimated values of NO release contents (Schoenfisch et al., *Anal. Chem.* **2013**, *85*, 1957).

Supplementary Fig. 37. Quantification of nitrite contents using the standard Griess assay by measuring UV-vis absorbance spectra (a) G1, (b) G3, (c) G5, and (d) BP1 micelles in the presence of Griess reagent under 630 nm irradiation. (e) UV-vis absorbance spectra of Griess reagent in the presence of varying amounts of nitrite. (f) Absorbance intensity at 545 nm as a function of nitrite concentrations.

- As shown in Supplementary Figs. 31-32, the authors mentioned the integration of the nanoparticles did not so changed under acidic conditions. This result would mean the positive charges depending on the protonation of TA moiety might occur only at the surface of the nanoparticle. The deprotonated form of TA moiety inside the nanoparticle may affect the liberation of NO from nanoparticles such as oxidation of TA by NO or so. The authors should consider and discuss the possibility of the interference of NO release from the nanoparticles under both neutral and acidic conditions.

Reply: We sincerely appreciate the reviewer's insightful comments and valuable suggestions. In response to the concerns raised, we have taken the following steps to investigate the potential interactions between tertiary amines and NO under both neutral and acidic conditions.

Firstly, we employed 2-(azepan-1-yl)ethan-1-ol as a model tertiary amine and exposed it to a saturated NO solution (1 equiv.) under both pH conditions. To monitor any chemical changes, we conducted NMR spectroscopy experiments. Interestingly, we observed no significant alterations in the NMR spectra, indicating that the reaction between tertiary amines and NO is unlikely to occur under either neutral or acidic pH conditions.

Furthermore, it is essential to emphasize that the NO concentration employed for these NMR studies was relatively high, at approximately 60 mM. In contrast, the NO released from our **G1** micellar nanoparticles was found to be substantially lower, typically below 0.1 mM. This observation strongly suggests that the tertiary amines present within the **G1** micellar nanoparticles exhibit a limited propensity to react with NO, as evidenced by their negligible response to the high NO concentrations used in our experiments.

Supplementary Fig. ^1H NMR spectra in $\text{DMSO-}d_6$ for 2-(azepan-1-yl)ethan-1-ol at (a) neutral and (b) acidic pH with or without saturated NO solution.

6. As for Fig. 3f, the authors should refer to the appropriate paper for NOFP, and also clarify the quantitative results of NO formation via NOFP, which would reflect the NO releasing efficiency in the biological conditions.

Reply: According to the review's suggestion, we have included reference 47 (Galindo et al. *Chem. Commun.* **2014**, 50, 3579) in the revised manuscript regarding the detection of NO using the NOFP probe. In addition, a standard calibration curve was generated and the amount of NO release was calculated to be $\sim 82.1 \mu\text{M}$ under 630 nm light irradiation for 30 min, corresponding to a yield of 66.6%. This result concurs well with the value ($\sim 69.2\%$) determined by UV-vis spectra (Table 1 and **Supplementary Fig. 36d**).

Supplementary Fig. 34. (a) The proposed working mechanism of NOFP. Evolution of fluorescence emission spectra of pyrylium cation-based NO probe (NOFP) ($50 \mu\text{M}$) under varying conditions: (b) **G1** micelles without light irradiation, (c) **G2** micelles under 630 nm light irradiation, and (d) **BP1** micelles under 630 nm light irradiation. (e) Emission spectra of NOFP in the presence of varying amounts of NO. (f) Standard calibration curve for NOFP at 525 nm.

Fluorescence intensity at 525 nm as a function of NO concentrations.

7. The authors should mention the methods for NO measurement such as ESP spin trapping measurement as well as NOFP.

Reply: As suggested by the reviewer, the detailed experimental procedures have been added (Supporting Information, pages S8-S9).

“EPR tests of NO release. EPR spectroscopy was recorded on a JES-FA200 (JEOL) spectrometer. The measurements were conducted at room temperature and the following parameters were used: modulation frequency: 100 kHz; modulation amplitude: 0.35 mT; scanning field: 324.3 ± 5 mT; microwave power: 1 mW; microwave frequency: 9.063 GHz. PTIO was used as a spin-trapping agent. The micelle concentration was 0.2 g/L containing PTIO(30 μ M) in all cases and the mixture was subjected to 630 nm light irradiation at pre-determined time intervals.”

“Monitoring NO release of micelles by NOFP. Micelles (0.2 g/L, PBS 7.4, 10 mM) were irradiated under 630 nm light (39 mW/cm²), and the irradiated dispersion (1 mL) was immediately mixed with NOFP solution (50 μ M). The mixture was stirred at room temperature for 5 min and the fluorescence intensities were measured.”

8. As for Fig. 3h, the authors should clarify and measurement methods for the oxygen concentration measurement.

Reply: As suggested by the reviewer, the detailed experimental procedures have been added (Supporting Information, pages S8).

“Dissolved oxygen measurement. The O₂ concentration measurements were acquired by an oxygen electrode (JPSJ-605F, INESA Scientific Instrument Co., Ltd). The micelles (2.5 mL) were respectively added into a tube, and the oxygen electrode was submerged in the micelle dispersion. Under 630 nm light irradiation, the concentrations of dissolved oxygen were continuously recorded.”

9. In accordance with Supplementary Figs. 31-32, the nanoparticles are still assembled under acidic conditions, which would mean that the TA moiety at the surface of the nanoparticle is only protonated. However, in Supplementary Fig. 37, the oxygen consumption was found to be decreased under acidic conditions at a significant extent, which may mean that a significant population of TA was protonated. This result would be not fully consistent with those suggested from Supplementary Figs. 31-32. The authors should explain and comment on these points.

Reply: We sincerely appreciate the reviewer’s insightful comments and the opportunity to clarify the observations in our manuscript. The **G1** micelles were formed through the co-assembly of two distinct diblock copolymers: the pH-inert **BP1** and the pH-sensitive **BP3**. Under acidic conditions, it is indeed

accurate to state that the TA moiety on the surface of the nanoparticle is primarily protonated. However, the seemingly inconsistent observation in **Supplementary Fig. 37** (now **Supplementary Fig. 38**), where oxygen consumption decreased under acidic conditions, can be explained by the unique properties of **BP3** diblock copolymers.

BP3 diblock copolymers, bearing tertiary amine moieties, exhibit an ultrasensitive pH-responsive behavior. This behavior allows them to maintain the micellar morphology without significant size changes even with a protonation degree of TA moieties ranging from approximately 0% to 90% (as previously reported, Gao et al., *Nat. Commun.* **2016**, *7*, 13214). This phenomenon contrasts conventional expectations that protonation of TA moieties at acidic pH should lead to swollen micelles with increased sizes.

Thus, the negligible size change observed in **Supplementary Figs. 31-32** does not necessarily imply that only the tertiary amines on the surface of **G1** micelles are protonated at pH 4.5. To further support this, we performed a titration of the pK_a values for **G1** micelles. Our calculations yielded pK_a values of 6.27 for **BP3** and 5.81 for **G1** micelles, influenced by the presence of **BP1** amphiphiles. These results indicate that approximately 2.5% of tertiary amine moieties remain in the non-protonated state at pH 4.5 within **G1** micellar nanoparticles. Notably, these non-protonated primary amines are still capable of scavenging 1O_2 , leading to the observed decrease in dissolved oxygen concentrations under 630 nm light irradiation, as shown in **Supplementary Fig. S38b**. We hope this clarification addresses your concerns regarding the protonation of TA moieties in **G1** micelles under acidic conditions.

Supplementary Fig. 38. (a) Changes in dissolved oxygen of aqueous dispersions of **G3-G6** micelles (0.2 g/L) in PBS (pH 7.4, 10 mM) under 630 nm light irradiation. (b) Changes in dissolved oxygen of aqueous dispersions of **G1** micelles (0.2 g/L) at pH 7.5, 5.5, 4.5, and **BP1** micelles in the presence or absence of

sodium ascorbate (10 mM) under 630 nm light irradiation (39 mW/cm²).

10. In general, the nanoparticles in this study seem larger in size than known antibacterial agents. The authors should explain whether these nanoparticles can penetrate biofilms practically even with such large size, or only attached on the surface of biofilms and degraded them from the outer-rim of biofilms.

Reply: Inspired by the review's suggestions, we have carefully examined previous publications regarding the penetration of micellar nanoparticles into biofilms. Indeed, previous results revealed that nanoparticles with a diameter of approximately 50 nm showed effective biofilm penetration and exhibited bactericidal properties (Rotello et al., *J. Am. Chem. Soc.* **2020**, *142*, 10723). In our work, the micellar nanoparticles had similar diameters and we further examined the biofilm penetration of these NO-releasing micellar nanoparticles by CLSM studies. To observe the micellar nanoparticles, we prepared rhodamine B (RhB)-labeled copolymers and fabricated **G8** micellar nanoparticles (similar to **G1** compositions with only replacing PdTBTBP with RhB), the enhanced biofilm penetration can be readily detected by the appearance of yellow emission (green for biofilm and red for **G8** micelles). In contrast, **G7** micelles without pH-responsive tertiary amine moieties showed no evident biofilm penetration. It is worth noting that the biofilms show no evident dispersal, which thus does not support the degradation of biofilms from the outer rims. Therefore, the incorporation of pH-responsive tertiary amines can not only render the resulting micellar nanoparticles adaptive to biofilm microenvironments but also enhance biofilm penetration (**Fig. 4b** and **Supplementary Fig. 45**). Collectively, the NO-releasing micelles (**G1**) can indeed penetrate biofilm rather than attaching onto biofilms, enabling photo-mediated NO release both at the periphery and inner layers of biofilm, which can thus be more efficiently eradicate biofilm infections.

Supplementary Fig. 45. (a) Representative 3D projection of image z-stacks showing the distribution of **BP6** and **BP8** micelles (red) in CRPA biofilms (Green). (b) Depth penetration in CRPA biofilms after treatment with **G7**, **G8**, and **BP8** micelles for 0.5 and 1 h (n = 10).

11. In accordance with Fig. 4b and Supplementary Fig. 44, the nanoparticles looked to access the artificial biofilm from its surface. Is the biofilm assembly affected with the penetration of the nanoparticles? If so, the oxygen concentration and pH values may almost same as that outside of the biofilm. The authors are expected to consider such possibility.

Reply: We are grateful to the reviewer for the astute observation, which prompted us to investigate the potential impact of **G8** micellar nanoparticles on the biofilm assembly and microenvironment. In response to this query, we conducted experiments employing an oxygen-sensitive probe, $\text{Ru}(\text{dpp})_3\text{Cl}_2$, to closely monitor changes in oxygen concentrations in the presence of **G8** micellar nanoparticles under dark conditions. Our experimental results, as presented in **Supplementary Figure 46**, demonstrate that whether subjected to **G8** micelle treatment or left untreated, the hypoxic gradient within the biofilm remains unaltered. This finding suggests that the penetration of **G8** micelles does not significantly affect the local microenvironments of biofilms. Consequently, the heterogeneous microenvironments within biofilms remain, making it an attractive challenge to develop antibiofilm agents capable of overcoming this inherent heterogeneity.

Supplementary Fig. 46. (a) 3D CLSM images of CRPA biofilms stained with an O_2 -specific probe ($\text{Ru}(\text{dpp})_3\text{Cl}_2$). The biofilm was treated with or without **G8** micelles (0.2 g/L). (b) Quantitative analysis of fluorescence intensities of the red channel.

12. As for the results in Fig. 5, the authors should refer to the appropriate papers for the biological effects of NO on the expression of the genes of drug resistance and biofilm formation. NO may affect the biofilms via chemical degradation or via inhibition of enzymes, in which the gene expression may rather increase through bacterial response to NO stimuli. The authors should consider and explain those possibilities.

Reply: Thanks for the reviewer's insightful suggestion and we have included some representative references regarding the biological effects of NO in the revised manuscript (Refs. 53-55). The revised manuscript is now read as follows (page 18, 1st paragraph):

“Notably, the NO release from G1 micelles led to significant downregulation of genes associated with multidrug resistance (e.g., mexC and mexD), the expression of these genes is closely associated with multidrug resistance efflux pumps, resulting in a significant reduction in CRPA drug resistance (Fig. 5b and Supplementary Fig. 51).¹⁴ In addition, the expression of genes related to biofilm formation (e.g., dgcA) was largely down-regulated (Fig. 5b,c)⁵⁰⁻⁵². It is noteworthy that the dgcA gene mediates the synthesis of DGC protein, which plays a crucial role in the synthesis of cyclic di-GMP (c-di-GMP). While c-di-GMP serves as a near-ubiquitous second messenger that governs a multitude of bacterial behaviors and holds pivotal significance in orchestrating the transition between motile loner cells and biofilm formers,⁵³ the reduction in c-di-GMP levels eventually induce the dispersion of biofilms^{54,55}.”

Reviewer #2:

This manuscript describes a micelle nanoparticle adapted to the heterogeneous biofilm microenvironment of drug-resistant bacteria that releases nitric oxide against the biofilm through two different photoredox catalytic mechanisms. The new material has potential implications for treating drug-resistant bacteria. In general, the manuscript content is very complete and comprehensive. Therefore, I propose to publish it with minor modifications. Here are some concerns:

Reply: We would like to express our heartfelt thanks to the reviewers for dedicating your valuable time to review our manuscript and for your positive comments on our research.

1: Table 1 provides the release of nitric oxide from G1 to G6. I suggest the authors show the release curve of nitric oxide, including micelles (G1-G6) and diblock copolymers (BP1-BP8).

Reply: According to the reviewer's thoughtful suggestion, we have presented the NO release curves for G1, G3, G5, and BP1 under 630 nm light irradiation (**Supplementary Fig. 36d**). The remaining groups (G2, G4, and G6), devoid of NO release moieties, function as controls in this study.

Supplementary Fig. 36. Evolution of UV-vis absorbance spectra of (a) **G3** and (b) **G5** micelles in PBS (pH 7.4, 10 mM) under 630 nm light irradiation. (c) **G1** micelles in PBS (pH 7.4, 10 mM) under dark conditions. (d) The nitric oxide release profiles of **G1**, **G3**, **G5**, and **BP1** micelles under 630 nm light irradiation (39 mW/cm²).

2: In Figure 4c, the fluorescence shown by the staining of the LIVE/DEAD bacterial viability kit is only one kind of fluorescence. Please provide the staining of live bacteria and the fluorescence of dead bacteria.

Reply: Thanks for the reviewer's thoughtful suggestion. We have provided the dual staining images in **Supplementary Fig. 47a**, indicating that only the treatment of **G1** micelles under 630 nm light irradiation can efficiently disperse biofilms and eradicate bacterial pathogens.

Supplementary Fig. 47. (a) 3D Confocal microscopy images of Live/Dead staining of CRPA biofilms treated with **G1**, **G2**, **BP1** micelles (0.2 g/L), and Cip (25 μg/mL), with or without 630 nm light irradiation for 30 min (39 mW/cm²). Green: live bacteria, Red: dead bacteria. (b) Quantification of biofilm thickness receiving various treatments.

3: It can be seen from Figure 4 that G1 has a good effect against biofilm, but the figure cannot reflect the dissipating biofilm effect of nitric oxide, whether it can provide the biofilm thickness after the materials act on the biofilm.

Reply: Inspired by the reviewer's suggestion, we quantitatively analyzed the biofilm thickness with various treatments by using ImageJ software, and the result is shown in **Supplementary Fig. 47b**. Only the **G1 +hv** group can lead to a significant decrease in biofilm thickness, while other treatments cannot efficiently

eradicate the biofilms. This observation underscores the notable efficacy of **G1** micelle in biofilm clearance under red light irradiation.

4: The histological and immunofluorescence analysis in Fig. 6f and Supplementary Fig. 52 Show the intensity of CD31 and TNF- α . Please explain why the intensity of CD31 rises first and then decreases but the intensity of TNF- α tends in the opposite trend.

Reply: We extend our appreciation to the reviewer. According to previous literature on wound healing (L. A. DiPietro, *Angiogenesis and wound repair: when enough is enough*, *J. Leukoc. Biol.* **2016**, *100*, 979), there is a rapid proliferation of new capillaries into the wound site during wound healing. The **G1** +hv treatment with photo-controlled NO release enhances the formation of blood vessels within wound tissues, resulting in an augmentation of CD31 intensity. Over time, most of the newly formed vessels regress or undergo selective apoptosis, eventually returning the blood vessel density to normal levels with decreased CD31 signals (Kraemer et al., *Neovascularization, Angiogenesis and Nutritive Perfusion in Wound Healing*, *Eur. Surg. Res.* 2018, *59*, 232). On the other hand, the production of TNF- α , an inflammatory cytokine, is produced in substantial quantities during the early stages of bacterial infection in the wound. However, the NO release by the **G1** + hv treatment exerts an anti-inflammatory effect, constantly lowering TNF- α levels during the entire healing process.

5: The material's view on heterogeneous biofilms is novel, but in the anti-biofilms and animal parts, please discuss the advantages of the material at different pH and oxygen concentrations.

Reply: Thank you for your insightful suggestion. We appreciate the opportunity to further elucidate the advantages of our micelles in different pH and oxygen concentration environments. Our designed micelles have been carefully engineered to adapt to the heterogeneity of biofilms. They possess unique features that enable them to function effectively under varying pH and oxygen conditions. Specifically, these micelles release nitric oxide (NO) both within and outside the biofilm, and their sensitivity to the biofilm environment allows them to excel in complex microenvironments. This adaptability ensures that they maintain their therapeutic efficacy even in challenging conditions. In an animal wound model with open wound biofilm infections, the wound area's complex microenvironment often presents challenges for biofilm infection-associated wound treatment (i.e., neutral pH/normoxia outside the biofilm and acidic pH/hypoxia within the biofilm). Our **G1** micelles have demonstrated the ability to release NO within the wound biofilm, which leads to biofilm dispersion and inflammation suppression. Importantly, this capacity extends to environments with varying pH and oxygen concentrations, making them highly versatile and effective therapeutic agents in a range of clinical scenarios. Overall, by functioning effectively under different pH and oxygen conditions, our micelles not only provide a novel approach to heterogeneous biofilm treatment but also offer promising clinical advantages for improved patient outcomes. We have included the above discussion in the revised manuscript.

Reviewer #3:

The noteworthy results are that NO release can be initiated from an N-nitrosoamine in the presence of red light with the goal to treat biofilm infections. The difference between this work and their previous work, published in JACS, is that in these studies, nanoparticles were produced by co-assembling diblock

polymers that contained the key delivery and release components: Pd-photocatalyst, tertiary amines and an *N*-nitrosoamine. The nanoparticles were charged to encourage penetration into the biofilm for optimal place for biofilm dispersion. While in the films, the light caused the activation of the photocatalyst which then led to NO production. The NO production levels were high enough to disperse the biofilm. The evidence for NO release is the decrease in the UV-signal of the functional group. Finally, the manuscript describes the effectiveness of these nanoparticles on biofilm eradication in an in vivo model. This is a very complete study from design through application. The measurements were all done in at least triplicate with the average and standard deviation reported. The results are confirmed by the evidence (appropriate characterization tools used). If there was one suggestion it would be to see if they could measure the NO itself in vitro through a selective sensor or chemiluminescence based method. While UV vis analysis of the product formed after analysis is of high value, the bio-useful amount of NO may also provide some insights into the utility of these materials for other applications as well. Indeed, this is a thoughtful and well designed study.

Reply: We would like to express our heartfelt thanks to the reviewers for dedicating your valuable time to review our manuscript and for your positive comments on our research. According to the reviewer's insightful suggestions, we employed the standard Griess assay to measure the NO release contents of these micelles under 630 nm light irradiation (**Supplementary Fig. 37**). Specifically, the NO release efficiency was determined to be 62.6 μM (50.8%), 16.7 μM (13.6%), and 25.6 μM (20.8%) for **G1**, **G3**, and **G5** with 30 min of light irradiation, respectively. It should be mentioned the NO-releasing efficiencies calculated by the Griess assay were lower than those determined by UV-vis spectrometry. This result was reasonable because the Griess assay can only estimate the nitrite concentration, leading to underestimated values of NO release (Schoenfish et al., *Anal. Chem.* **2013**, *85*, 1957–1963). Please also kindly refer to our replies to **Reviewer 1** in point 4.

REVIEWERS' COMMENTS

Reviewer #1 (Remarks to the Author):

In the revised manuscript, the authors responded to all the reviewers' comments, and revised their manuscript in accordance with them. It seems largely improved but there is an ambiguous point. As for Supplementary Fig. 38, is this system under closed conditions, or open air? In the practical situation, the rate of oxygen diffusion would be highly depending on the biofilm conditions. If the rate of oxygen diffusion is high, the hypoxic conditions, which is required for efficient NO formation, would not be efficiently achieved. The authors should clearly discuss the practical possibility of hypoxic conditions under open air circumstance.

Reviewer #2 (Remarks to the Author):

All I concerned have been revised and responded, so I think it could be accepted now.

Reviewer #3 (Remarks to the Author):

The authors have responded to the reviews in an appropriate manner.

REVIEWERS' COMMENTS

Reviewer #1 (Remarks to the Author):

In the revised manuscript, the authors responded to all the reviewers' comments, and revised their manuscript in accordance with them. It seems largely improved but there is an ambiguous point.

As for Supplementary Fig. 38, is this system under closed conditions, or open air? In the practical situation, the rate of oxygen diffusion would be highly depending on the biofilm conditions. If the rate of oxygen diffusion is high, the hypoxic conditions, which is required for efficient NO formation, would not be efficiently achieved. The authors should clearly discuss the practical possibility of hypoxic conditions under open air circumstance.

Reply: We appreciate the insightful comments provided by the reviewer. The reviewer is definitely right the local oxygen concentration within biofilms is highly dependent on oxygen diffusion rates. Changes in dissolved oxygen concentrations were tested in open air in Supplementary Fig. 38, and this information has been added to the legend and the main text of the manuscript. This result suggests that our photoredox catalysis platform can be successfully operated in aerated conditions even in the presence of a high oxygen concentration, which was beneficial for overcoming the oxygen heterogeneity in infection wounds. We have included the result and discussion in the revised manuscript (page 10).

“We observed a significant reduction in dissolved oxygen levels, from ~ 8.1 to ~ 0.02 mg/L within 5 min for G1 and G2 micelles under 630 nm light irradiation at pH 7.4 under open air conditions (Fig. 3h). This finding underscored the successful implementation of the photoredox catalysis process without the need for additional oxygen-scavenging agents, rendering it advantageous for potential applications in pathological tissues characterized by intricate oxygen gradients.”